# The Influence of Heat Treatment on Corrosion Resistance and Microhardness of Hot-Dip Zinc Coating Deposited on Steel Bolts

**DOI:** 10.3390/ma15175887

**Published:** 2022-08-26

**Authors:** Dariusz Jędrzejczyk, Elżbieta Szatkowska

**Affiliations:** 1Department of Mechanical Engineering Fundamentals, University of Bielsko-Biala, Willowa 2, 43-309 Bielsko-Biała, Poland; 2BOSMAL Automotive Research and Development Institute Ltd., Sarni Stok 93, 43-300 Bielsko-Biala, Poland

**Keywords:** heat treatment, corrosion resistance, hot-dip zinc galvanizing, coating hardness

## Abstract

The analyzed topic refers to the corrosion resistance and changes in microhardness of the heat-treated (HT) hot-dip zinc coating deposited on bolts. The research aimed to evaluate the influence of the HT on the increase of the coating hardness and changes in anticorrosion properties. Hot-dip zinc coating was deposited in industrial conditions (acc. EN ISO 10684) on chosen bolts (M12x60). The achieved results were assessed based on corrosion resistance tests in neutral salt spray (salt chamber) and microhardness measurements. Tests were conducted in accordance with the adopted fractional plan, generated in the DOE module of the Statistica software. Using the conjugate gradient method optimal parameters of HT were determined. The conducted tests proved that the controlled heat treatment may increase the hardness of the hot-dip zinc coating without a significant deterioration in its basic protective function (corrosion resistance). The observed changes in the hardness and corrosion resistance of the zinc coating are a consequence of changes in its structure.

## 1. Introduction

Zinc is the most commonly used element in the production of anticorrosion coatings [1]. Processes of Zn coating deposition are relatively simple and do not require advanced devices, complicated technologies or large financial resources [2]. The report of the International Lead and Zinc Study Group regarding years 2016–2020 [3] states that global zinc production amounted to about 12 million tons yearly and more than 6 million tons were used as anticorrosion protection for steel. In practice, four galvanizing methods are used. The oldest method used on an industrial scale for corrosion protection of steel parts is immersion galvanizing, commonly known as “hot-dip galvanizing”. By contrast, the latest technology using zinc to protect steel surfaces is the so-called “lamella technique”. This is a typically adhesive process—the surface before immersion is prepared to obtain the greatest possible degree of development. In addition to the methods mentioned above, sherardization (where the coating has a structure similar to hot-dip galvanizing) and electro-galvanizing are also applied [4,5,6].

The most commonly used method for protecting industrial steel parts is hot-dip galvanizing [7,8,9]. Because hot-dip zinc galvanizing ensures high quality and long-term protection against corrosion, examples of its application can be found in every environment (marine, rural, industrial) and different kinds of industry (shipbuilding, machine, agriculture)—in most environments corrosion protection lasts on average 25–75 years, depending on the thickness of the coating correlated with the environment of exposure to corrosive agents [7].

Fasteners are protected primarily by hot-dip zinc galvanizing [5]—the thickness of the deposited coatings is usually in the range of 45–65 μm [4], and the coating microstructure has a typical diffusive character and consists of several intermetallic phases of the Fe–Zn system (η, ζ, δ, Γ) [10]. In addition to hot-dip galvanizing, electro-galvanizing is also used for anticorrosion protection for different kinds of bolts, especially those with a smaller diameter. However, the coating obtained by this method is usually much thinner, which usually results in lower corrosion resistance. This process also entails the risk of reducing the mechanical properties of the base material due to hydrogen embrittlement [11] and has a negative impact on the environment [12]. To achieve a thinner zinc coating on the surface of bolts characterized by comparable corrosion resistance to the standard hot-dip zinc galvanizing, the high-temperature galvanized method is applied. As a result of this process, the zinc coating is also composed of Fe–Zn intermetallic phases. Since galvanizing takes place at a temperature about 100 °C higher than during standard hot-dip galvanizing, the process requires the use of ceramic or ceramic lining baths (instead of the steel ones used in most galvanizing plants), which results in higher costs of energy and equipment [13]. In addition, in the case of galvanizing bolts made in the higher strength classes, the high-temperature method carries the risk of deterioration of the bolts’ strength properties as a result of tempering [4]. A coating with a similar structure, as in the case of high-temperature galvanizing (composed of intermetallic phases), is also deposited using thermo-diffusion galvanizing [14]. As a result, parts showing very good anti-corrosion protection, without hydrogen embrittlement [15], are obtained. However, in comparison to hot-dip galvanizing, this method has limitations related to the size of the parts which can be subjected to this process (it is not possible to galvanize parts as large as those in the case of hot-dip zinc coating in galvanizing baths) and zinc plating takes more time (several, or up to even a dozen, hours).

One of the weaknesses of the hot-dip zinc coating is the relatively low hardness. The hardness measured on the zinc surface of the hot-dip coating, i.e., the hardness of the η layer, according to various sources oscillates around the value of 50 HV (50–57 HV [16]; 52 HV [17]; 41 HV [18]). One of the methods of improving the properties of commonly used coatings is heat treatment. In the literature on materials subjected to elevated temperatures, the attention is focused mainly on oxidation mechanisms [19,20,21,22], influence of different elements on the oxidation resistance [23,24] and the mechanical properties of oxidized materials [25]. In addition to the above, the subject of the coatings’ controlled heat treatment and its impact on the properties of coatings is also increasingly taken up by researchers testing various types of coatings (Zn-Ni-P-W [26], Ni-W [27]; diamond-like [28]; bronze [29]; Ni-B [30]). In the presented article, the HT impact on the structure of coatings and their key properties related to exploitation conditions are analyzed.

Azadeh et al. [31] analyzed the effect of heat treatment on the deformability of hot-dip galvanized DC01 steel. Samples were subjected to temperature exposures of: 500, 510, 520, 530 and 540 °C. The treatment time was from 10 to 180 s. After the heat treatment, the samples were cooled in water. The presented results of the research (SEM, EDS, XRD, FLD) showed that the applied parameters (temperature and time) of the treatment result in structure changes in the coating. It was found that the use of higher temperatures in combination with shorter processing times allows for improving the plasticity of the zinc coating.

In the research of Szabadi et al. [32], the subject of which was the abrasive wear of the hot-dip zinc coating applied to S235JRG2 steel, the results of microhardness measurements using the Vickers method as well as tests of the condition and qualitative composition of the surface are presented (SEM-EDS). The authors compared data obtained for the samples galvanized in the crude state and the ones galvanized after being subjected to heat treatment. Measurements of abrasive wear were conducted in a specially designed abrasion tester. The researchers claim that the heat treatment caused a change in the structure of the zinc coating, which in turn increased abrasion resistance. The suggested reason for increased abrasion resistance is the intensification of iron diffusion. The zinc coating on the base sample had an average hardness value of about 47 HV. The heat treatment resulted in an increase in the hardness of the zinc coating to about 106 HV. In the work, however, there is no detailed data on the basic parameters of the heat treatment (time, temperature).

In the work of P. P. Chung et al. [33], the authors note the phenomenon of the beneficial effect of heat treatment on zinc coatings applied to fasteners. The research concerned zinc coating deposited onto 1022 steel bolts and flat samples made of CA3SN-G steel. The researchers heat-treated the galvanized parts using a temperature of 340 °C and a time of 10, 15 and 30 min. Heat treatment was carried out in ambient air. Corrosion resistance was tested by the potentiodynamic method in a 5% NaCl solution. Microscopic examinations (optical LOM, scanning SEM with EDS analysis) and structure analysis (XRD) were also carried out. The authors claim that the heat treatment caused changes in the structure of intermetallic phases, which contributed to the improvement of corrosion resistance because they could become a barrier to progressive corrosion processes. The researchers also observed an increase in the thickness of the treated coatings in relation to samples that were not heated at 340 °C.

So, considering the above analysis and the opinion of authors of the article [34], we can state that corrosion resistance is the most important property of zinc coating. How zinc protects iron alloys (steel, cast iron) against corrosion depends on its specific physico-chemical properties [35]: in relation to iron, zinc becomes the anode that is first attacked by corrosion; zinc can create a corrosion resistance barrier and shows a high passivation ability.

The anti-corrosion properties of hot-dip zinc coating depend on many factors, including microstructure [36,37] and thickness [37,38,39,40]. There is no absolute uniqueness among data regarding the corrosion resistance of individual layers of zinc coating. There is also a statement in the literature [33,41] confirming that corrosion rate in alloyed zone is usually lower and thicker coatings exhibit higher corrosion resistance due to the presence of a larger fraction of Fe–Zn phases. However, most publications confirm the lower corrosion resistance of alloyed layers in comparison to pure zinc η layer in different environment—both in fresh, hardened and chloride-contaminated concrete [42] and in wet–dry cyclic conditions, in chloride-containing environments [43]. In article [44], authors claim that the low-temperature annealing (225 °C) causes the formation of intermetallic Fe/Zn compounds, a transformation of amorphous oxide inclusions to the crystalline form and a decrease in the Zn lattice parameter for Zn–Co and Zn–Fe alloys, which also results in a decrease in corrosion resistance.

As shown in articles [45,46], an increase in HT temperature results in an increase in the thickness of the iron-rich layers (ζ, δ, Γ) in the zinc coating, and after the HT at 460 °C there is practically no pure η phase [45].

Considering the data from the literature and results of our own research presented in the previous article [47], where both disk samples and bolts were tested and the effect of heat treatment on the change of zinc coating microhardness, microstructure and coefficient of friction was studied, it seems crucial to assess the impact of the proposed heat treatment on the change in corrosion resistance of the tested coatings.

The aim of the presented research was to verify the possibility of increasing the hardness of zinc hot-dip coating by HT (up to 130–150 HV), with an acceptable reduction in corrosion resistance not exceeding 10%. During the experiment the impact of the heat treatment on the zinc coating deposited on typical steel bolts (M12x60) was assessed. Changes in microhardness (HV 0.02) values, microstructure character and corrosion resistance (NSS test) allowed to choose the optimal treatment conditions. The research was conducted in accordance with the adopted fractional plan, generated in the DOE module of the Statistica software. The optimal heat treatment parameters (ensuring hardness at the level of 140 HV.02 and corrosion resistance reduced by no more than 10%) were determined by the conjugate gradient method. The conducted studies confirmed that the controlled heat treatment allows a significant increase in the hardness of the zinc coating deposited on the tested bolts without a serious deterioration in the key property of the coating, i.e., corrosion resistance.

## 2. Materials and Methods

Steel bolts M12x60—8.8U (23MnB4 steel) were selected for the tests. On the one hand, the selected bolt diameter facilitated hot-dip zinc galvanizing (reducing the risk of zinc flashes formed on the thread). On the other hand, it met the requirements of the method used in the previous article for the friction coefficient determination (Schatz Analyze M12 testing machine system—where bolts with a maximum diameter of 12 mm can be tested). Samples were hot-dip galvanized according to EN ISO 10684 [48]—etching in 12% HCl, fluxing, dipping in Zn bath with Al, Bi and Ni at temp. of 460 °C within the time of 1.5 min and cooling in water.

Heat treatment was carried out in temperature range t = 270–430 °C in an electric chamber furnace. The bolts were held at the treatment temperature in time τ = 7–11 min. The parameters of heat treatment were selected based on the preliminary tests performed at work [47] with the use of disc-shaped samples measuring 25 mm in diameter and 4 mm in thickness and compared with the kinetics of heating disc samples and bolts (M12x60) using a thermal imaging camera. Table 1 presents values of the heat treatment parameters applied in the tests together with the variant numbers (1–9), i.e., the trivalent fractional plan (three values of variable parameters) for two input quantities (temperature and time of heat treatment). The plan was generated in the DOE (Design of Experiments) module of the Statistica software.

After the heat treatment samples were taken out of the furnace chamber and were air-cooled to the ambient temperature. The following parameters were analyzed during investigations: the heating process (chamber furnace FCF 75HM, Flir E95—thermal imaging camera), the microstructure of zinc coating structure and steel—with the use of a scanning microscope with EDS analysis (scanning electron microscope EVO 25 MA Zeiss with an EDS attachment), the microhardness changes measured on the bolts head (Vicker’s HV 0.02, Mitutoyo Micro-Vickers HM-210A device 810–401 D), corrosion resistance (salt chamber—Liebisch Labortechnik, type S 1000 M-TR, with capacity 1 000 l). To analyze the microstructure, chemical composition, phase identification in thin coatings, advanced devices (X-ray diffraction (XRD) and scanning electronic microscope (SEM) with EDS) were usually used [49,50]. In this study, to identify individual phases occurring in the zinc coating, only a scanning microscope with an EDS attachment was used, which seems to be quite sufficient due to the very characteristic and at the same time diverse morphology of the analyzed phases (η, ζ, δ, Γ).

Considering the planned effects of the heat treatment (coating hardness increase to the value in the range: 130–150 HV and acceptable reduction in corrosion resistance less than 10%), the optimal heat treatment parameters were determined using the conjugate gradient method [51].

## 3. Results and Discussion

### 3.1. Coating Thickness and Microhardness Measurements

Measurements of the thickness of the coatings deposited on galvanized bolts M12 × 60—8.8U were made by the magnetic method, in accordance with PN-EN ISO 2178:2016-06 [52]. The measurements were carried out in places defined by standard on the bolts heads before and after their heat treatment, using an electronic coating thickness gauge. On each bolt, 10 measurements were made, the results of which were averaged. For every bolt, the average coating thickness was in the range of 59.76 ± 1.27 μm, which corresponds to the requirements of the fasteners [4]. It follows from the achieved results that the applied heat treatment does not affect the change in the thickness of the zinc coating.

Microhardness measurements were carried out on the bolt’s head (outer layer) using the Vickers HV 0.02 method. Five measurements were made on the surface of each of the bolts. The obtained averaged results are shown in Figure 1.

Heat treatment conducted at fixed values (t = 270 °C, τ = 9 min) caused almost a twofold increase in the microhardness of the outer zinc coating layer. The highest—almost fourfold increase in microhardness (in comparison to the microhardness of coatings deposited on base bolts, without HT)—was obtained for coatings applied to bolts subjected to the treatment at the highest temperature and the longest time (t = 430 °C, τ = 11 min). Comparing the hardness measured after heat treatment at 430 °C and time τ = 11 min) (variant 9—Table 1) to the literature data (Table 2), it can be concluded that a phase *ζ* already appeared in the outer layer of the coating. Classical hot-dip zinc coating is composed of four phases according to the Fe–Zn diagram [17,53,54]: Г—Fe_3_Zn_10_, δ—FeZn_7_, ζ—FeZn_13_ and an iron solid solution in zinc—η (Figure 2a), which is settled on the surface during pulling out of the bath. After heat treatment carried out using parameters corresponding to the other variants (2–8), the outer layer of the coating was composed of a mixture of the phases η and ζ. The above statement was additionally verified on the basis of the microscopic observations and EDS analysis—Figure 2, the results of which are been related to the literature data—Table 3.

Figure 2b,d shows a view of the outer surface of the zinc coatings obtained under extreme conditions: without heat treatment (Figure 2b and after HT), using the longest time and the highest temperature (τ = 11 min., t = 430 °C; Figure 2d). Iron content was determined by EDS analysis in five randomly selected places on the outer surface of the zinc coating and the obtained results were averaged. The error of a single measurement did not exceed 5%. An increase in HT temperature results in an increase in the thickness of the iron-rich layers (ζ, δ)—the measured Fe content in areas on the outer surface (Figure 2b,d) was equal accordingly to 0.17 (standard deviation σ = 0.02) and 7.31% (σ = 0.27). Additionally, the linear EDS analysis confirmed the extension of the ζ, δ, Γ phases range in comparison to zinc coating without HT. The outer surface of the coating without heat treatment is composed of solid solution η (smooth and compact), while the surface after heat treatment is composed of a column-crystalline phase ζ (more diverse, heterogeneous, less compact). The conditions for zinc coating structure recomposition during heat treatment are completely different than for hot-dip zinc coating deposition during galvanizing—the Zn amount in the coating is limited. It is quite possible that during the HT, the liquid solutions diffuse along the channels between the cells of ζ and δ phases (like in the model presented by Wołczyński [60]), but a more probable is the occurrence of bulk diffusion that results in the thickening of ζ and δ phases, which in the extreme cases leads to the occurrence of one phase ζ on the outer surface of the coating (Figure 2c).

The analysis of the obtained results using the DOE module of the Statistica software made it possible to present the impact of heat treatment parameters on the change in the hardness of the applied coatings in the entire range of used parameters both in the form of a graph (Figure 3) and an equation (Equation (1)). The coefficient of determination (fitting equation to experimental data) for microhardness results is $R^{2}$ = 0.99.
(1)$${\hat{y}}_{HV}=-192.39+0.716t-0.000416t^{2}+18.3\tau -0.2916\tau^{2}$$

### 3.2. Corrosion Resistance Measurements

Both the bolts without HT and those that have been treated in accordance with the fractional experiment plan were placed inside the salt chamber. Samples were placed on a support made of chemically inert plastic in relation to the tested objects, allowing them to be positioned at an angle of ~20° from the vertical (Figure 4a) in accordance with the requirements presented in standard ISO 9227:2017 [61]. Five bolts treated at each of the selected temperatures and times were placed in the chamber. The appearance of red corrosion signs was controlled at intervals of 24 h.

Bolts were subjected to an accelerated corrosion resistance test in a neutral salt spray (NSS). The parameters of accelerated corrosion tests in the salt chamber were as follows: 5% NaCl, condensate pH 6.8–7.0, temperature 35 ± 1 °C, salt spray fall rate 1.5 mL/h. Figure 4b shows an example of the appearance of bolts after the corrosion resistance test. The achieved results are presented in Table 4. The bolts without the heat treatment were characterized by corrosion resistance of 360 h. This result was a reference point during the calculation of the reduction of corrosion resistance as a result of the conducted heat treatment (100% corresponds to the corrosion resistance of the bolts without HT).

Changes in the corrosion resistance expressed as a percentage reduction of the starting value are presented in Figure 5. The same resistance as the one determined for bolts without heat treatment was shown by the bolts treated in the following conditions: t = 270 °C, τ = 7 min; t = 270 °C, τ = 9 min; and t = 350 °C, τ = 7 min. In the case of bolts treated under the conditions as follows: t = 270 °C, τ = 11 min; t = 350 °C, τ = 9 min; t = 350 °C, τ = 11 min; and t = 430 °C, τ = 7 min, corrosion resistance decreased by 24 h (i.e., 6.6%). A reduction in corrosion resistance by 48 h, corresponding to 13.3%, was observed for bolts subjected to heat treatment under the following temperature and time conditions: t = 350 °C, τ = 11 min and t = 430 °C, τ = 9 min. In turn, the shortest time to the appearance of red corrosion was measured for bolts heat-treated under conditions: t = 430 °C, τ = 11 min, i.e., there was a reduction in resistance time under corrosive conditions by 72 h in comparison to bolts that were not heat treated (so the resistance decreased by 20%). Therefore, it can be concluded that the heat treatment carried out at the highest temperature in combination with the longest holding time results in the greatest loss of corrosion resistance. The reason for the decrease in corrosion resistance of the coating after heat treatment may be explained by the expansion of the range of occurrence of iron-enriched phases (which, according to some researchers, are characterized by lower corrosion resistance [33,41]), and in the extreme case, the appearance on the surface of only the phase ζ, which is less homogeneous and less compact than the phase η. As a result, surface development and surface roughness also increase. On the other hand, heat treatment can also lead to the appearance of microcracks visible in the cross-section of the coating (reducing the compactness). The obtained results are similar to those achieved in the previous tests with application of disk samples [47]. This also proves that the conducted analysis of heating kinetics was correct, and the heat treatment parameters selected on its basis made it possible to obtain comparable results on both flat samples and M12x60 bolts.

The analysis of the obtained results using the DOE module of the Statistica software made it possible to present the impact of heat treatment parameters on the change in the corrosion resistance of the applied coatings in the entire range of used parameters both in the form of a graph (Figure 6) and an equation (Equation (2)).
(2)$${\hat{y}}_{NSS}=358.60+0.18749T-0.0006249T^{2}+7.9\tau -0.9\tau^{2}$$

The coefficient of determination (fitting equation to experimental data) for corrosion resistance results is R^2^ = 0.95. Because the heat treatment has the opposite influence on the changes in hardness and corrosion resistance of the tested zinc coatings, the analyzed issue is an excellent area for the application of optimization solutions.

### 3.3. Solving the Optimization Problem

Regression equations determined on the basis of data obtained as a result of experimental studies (Figure 3 and Figure 6, Equations (1) and (2)) were used to formulate the optimization problem. In the conducted experiments, the influence of two parameters of the heat treatment was investigated (decision variables), i.e., temperature t and time τ, on microhardness, expressed in HV scale (criterion *q*_1_), and the corrosion resistance, expressed in hours of duration of the corrosion resistance test (criterion *q*_2_). The optimization problem was therefore formulated as a two-criteria task with component criteria *q*_1_ and *q*_2_.

To solve the above task, the concept of target programming was used, according to which the original two-criteria task is replaced by a single-criteria task, in which the cumulative criterion is the distance from the ideal point in the criteria space $\tilde{Q}\;\left({\tilde{q}}_{1},\;{\tilde{q}}_{2}\right)$ in the sense of the square of the Euclid metric [62]. The following values were accepted as the coordinates of the ideal point: ${\tilde{q}}_{1}$ = 140 HV i ${\tilde{q}}_{2}$ = 324 h. In the first step, in the space of decision variables, a set of acceptable solutions Φ was defined, which is determined by the permissible intervals of variation of heat treatment parameters (t = 270–430 °C, τ = 7–11 min) fixed on the basis of the analysis of the results of preliminary tests [47]. The permissible set took the shape of a rectangle shown in Figure 7. Having regression Equations (1) and (2), the task was transformed into a criteria space mapping the permissible set Φ in a set of permissible values of criteria Ѱ (Figure 8).

Each point in the set Φ was assigned a point $q={\left[q_{1}\left(x\right),\;\;q_{2}\left(x\right)\right]}^{T}$, adopting microhardness as $q_{1}=q_{1}\;\left(t,\tau \right)$ and corrosion resistance as $q_{2}=q_{2}\;\left(t,\tau \right)$. Then, due to the formulated assumptions regarding the postulated value of microhardness and the postulated reduction of corrosion resistance, the ideal point was determined, $\tilde{Q}={\left[{\tilde{q}}_{1},\;{\tilde{q}}_{2}\right]}^{T}$. Taking into account the thesis put forward in the work, the postulated value of microhardness (${\tilde{q}}_{1}$) should be in the range of 130–150 HV, and the ideal value of this criterion was the average value ${\tilde{q}}_{1}$ = 140 HV. It was further assumed that the postulated permissible reduction of corrosion resistance should not exceed 10% of the total corrosion resistance of the coatings applied to bolts without heat treatment (360 h). This value was accepted as a reference point and gave the ideal value of the criterion of ${\tilde{q}}_{2}$ = 324 h. Assuming that both component criteria are equally valid, the aggregate criterion expressing the distance from the ideal point takes the form as follows:(3)$$Q={\left({\tilde{q}}_{1}-q_{1}\right)}^{2}+{\left({\tilde{q}}_{2}-q_{2}\right)}^{2}\to min\;$$

The solution to the task of bi-criteria optimization in the goal space is the point $\hat{Q}$ belonging to a set of permissible values of criteria Ѱ located the closest to the ideal point $\tilde{Q}$. This point has coordinates $q_{1}$ = 142 HV and $q_{2}$ = 329 h (Figure 8). The exact coordinates of the optimal solution were achieved using conjugate gradient method [51,63].

After the inverse transformation of the task from the criteria space to the decision variable space using regression equations (Equations (1) and (2)), optimal point $\hat{Q}$ has the coordinates: temperature 280 °C and time 11 min (Figure 7), which corresponds to optimal hardness, $q_{1}$ = 142 HV, and optimal corrosion resistance, $q_{2}$ = 329 h (the duration of the test until the appearance of red corrosion).

## 4. Conclusions

(1)Microhardness measured on the heads of bolts subjected to the HT in the tested temperature range showed max. 4 times increase in the HV in the outer surface layer (85–204 HV 0.02) in relation to the reference samples (52 HV).(2)The observed changes in the hardness and corrosion resistance of the zinc coating are a consequence of changes in its structure. As the treatment time and temperature increase, the range of occurrence of the harder phase ζ increases too, at the expense of the η phase range (the total thickness of the coating does not change).(3)Only after a heat treatment conducted at 270 °C and a duration of 7 and 9 min and a heat treatment at 350 °C and a duration of 7 min, the corrosion resistance of the tested coatings did not decrease concerning coatings without heat treatment.(4)The obtained results confirm that it is possible to increase the hardness of the tested zinc coatings deposited on the bolts to a value of approx. 120 HV without reducing their corrosion resistance.(5)The obtained solution of the two-criteria optimization task allows for determining the most favorable/optimal parameters of heat treatment of hot-dip zinc coating (temperature and time) with reference to the postulated values of microhardness and corrosion resistance.(6)The determined optimal parameters of for the heat treatment of bolts (fulfilled the assumptions formulated in the work, i.e., HV = 140; corrosion resistance reduction ≤10%) in the tested range of variability of parameters are as follows: temperature 280 °C and time 11 min. The presented values of heat treatment parameters allow obtaining a coating hardness of 141 HV and coating corrosion resistance of 329 h.

## Figures and Tables

**Figure 1 materials-15-05887-f001:**
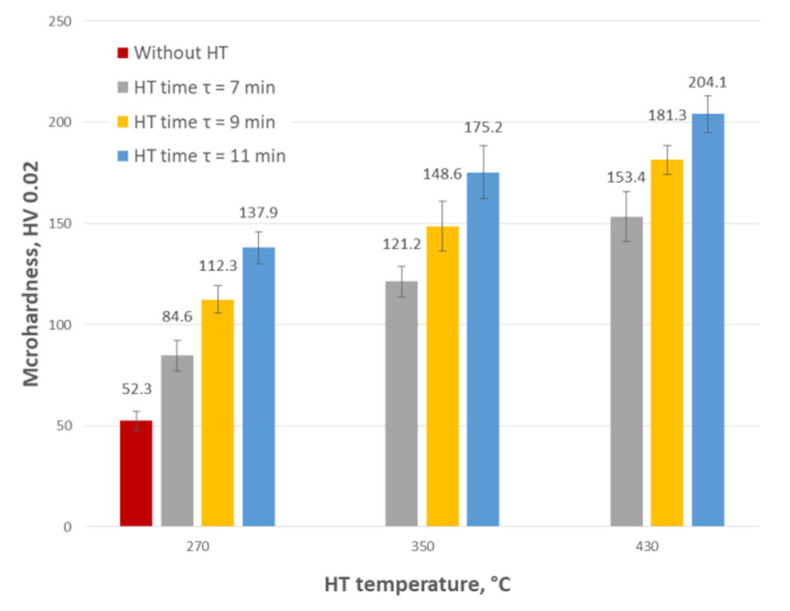
Comparison of the microhardness of zinc coatings measured on bolts surface before and after heat treatment.

**Figure 2 materials-15-05887-f002:**
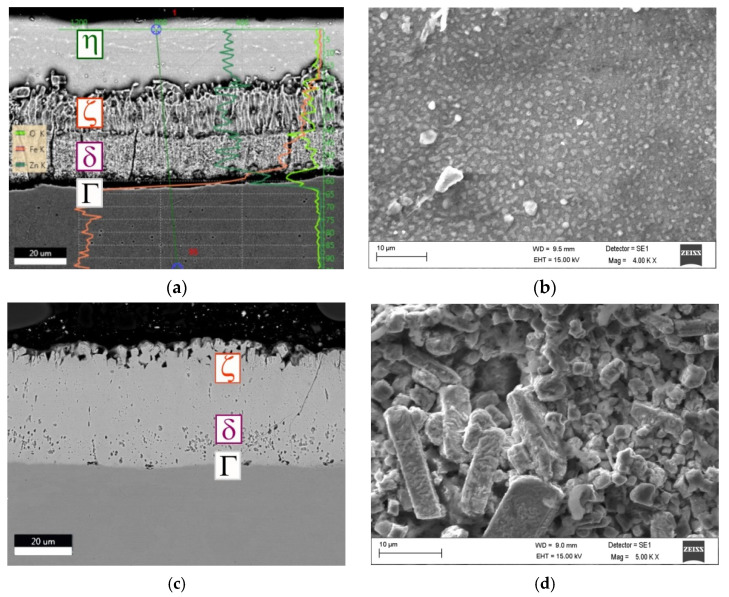
The microstructure observed in the cross section (**a**,**c**) and on the outer surface of zinc coating deposited on heads bolts (**b**,**d**) ((**a**,**b**)—without HT; (**c**,**d**)—t = 430 °C, τ = 11 min).

**Figure 3 materials-15-05887-f003:**
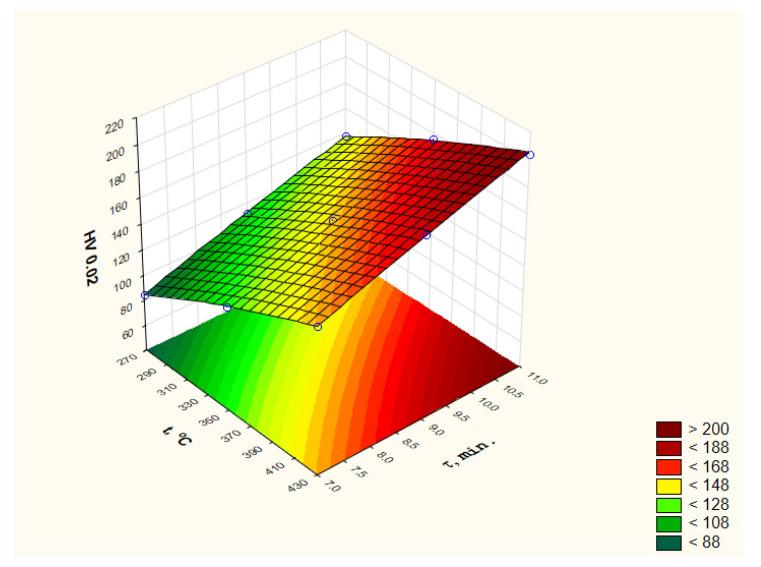
Influence of the heat treatment parameters on the microhardness of the zinc coating deposited on the head of M12 × 60 bolts—determined with “Statistica” software.

**Figure 4 materials-15-05887-f004:**
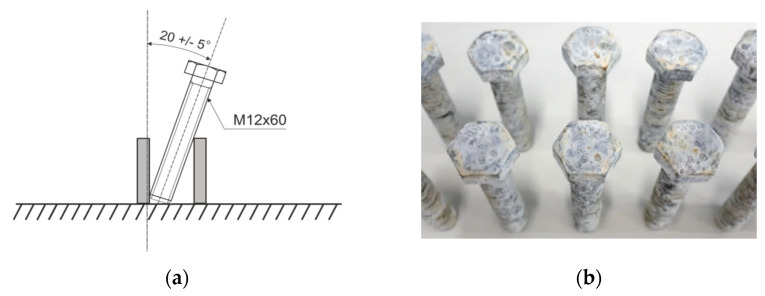
Bolt position during the NSS test (**a**) and their appearance after the test (**b**).

**Figure 5 materials-15-05887-f005:**
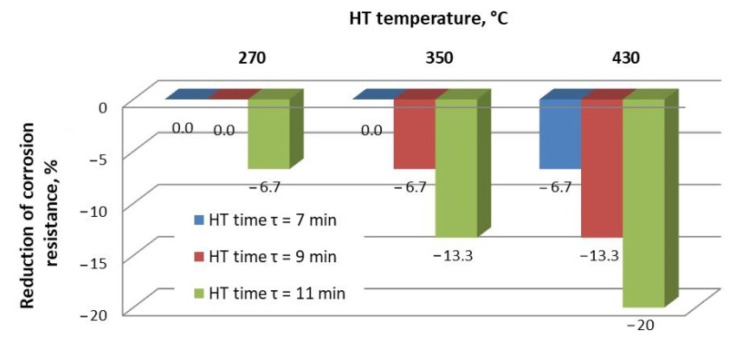
Changes in the corrosion resistance as a result of heat treatment expressed as a percentage reduction of the starting value.

**Figure 6 materials-15-05887-f006:**
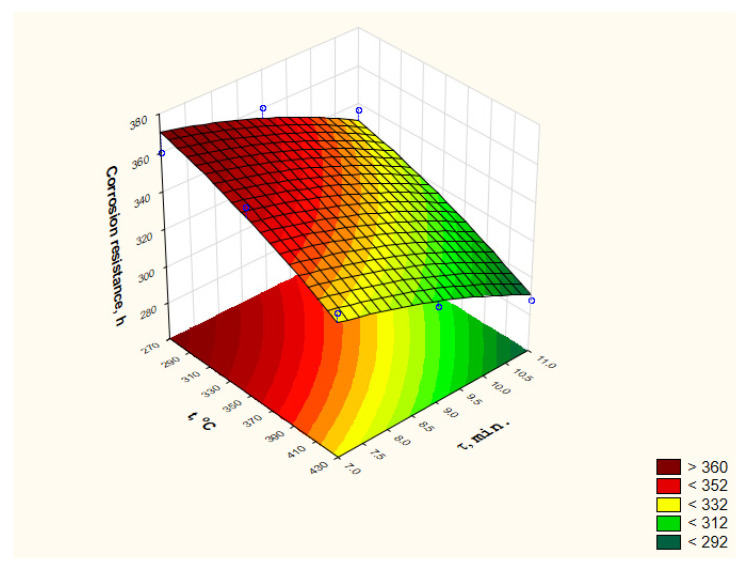
Influence of the heat treatment parameters on the corrosion resistance of the zinc coating deposited on the f M12 x 60 bolts—determined in “Statistica” software.

**Figure 7 materials-15-05887-f007:**
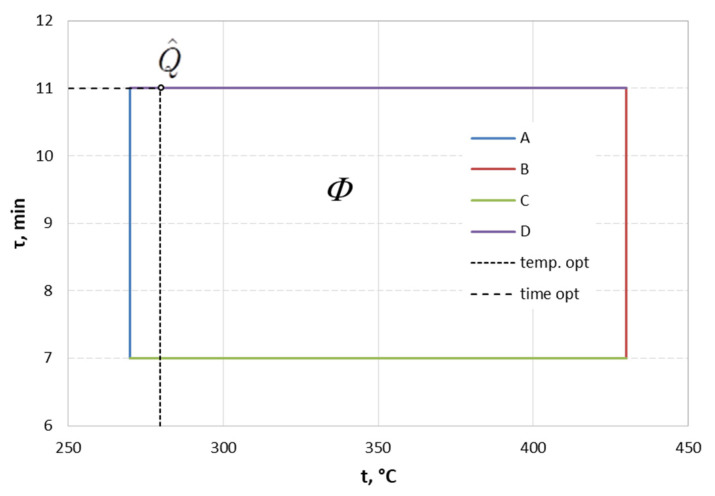
Set of acceptable HT parameters.

**Figure 8 materials-15-05887-f008:**
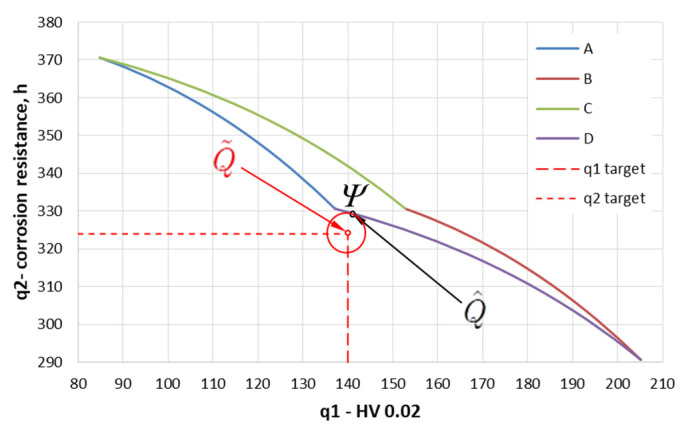
Set of acceptable criteria values.

**Table 1 materials-15-05887-t001:** Values of the heat treatment parameters and numbers of fractional plan variant.

Parameter of Heat Treatment		No. of Fractional Plan Variant
Temperature t, °C	Time τ, min.	
270	7	1
	9	2
	11	3
350	7	4
	9	5
	11	6
430	7	7
	9	8
	11	9

**Table 2 materials-15-05887-t002:** Hardness of individual intermetallic phases of zinc coating deposited by hot-dip zinc galvanizing according to: Evans [8] (**a**) and Pokorný et al. [55] (**b**).

Phase Kind		Phase Hardness	
		(a)	(b)
**Г**	Г, Fe_3_Zn_10_	326 HB	326
	Г_1_, Fe_5_Zn_21_	-	505 HV
**δ**	δ, FeZn_10_	-	358 HV
	δ, FeZn_7_	270 HB	-
**ζ**	ζ, FeZn_13_	220 HB	208 HV
**η**	η, Zn(Fe)	70 HB	52 HV

**Table 3 materials-15-05887-t003:** The content of iron in individual intermetallic phases of the zinc coating, deposited by ho-dip zinc galvanizing, according to: J. D. Culcasi et al. [54] (**a**), D. Kopyciński et al. [56] (**b**) and H. Kania et al. [57] (**c**).

Phase Kind	Content Fe,%	Phase Kind	Content Fe, %	Phase Kind	Content Fe, %
(a)		(b)		(c)	
**Г, Fe_3_Zn_10_**	18–31	Г_1_, Fe_3_Zn_10_	21–28	Г, Fe_3_Zn_10_	23.7–31.5
**Г_1_, Fe_5_Zn_21_**	19–24				
**δ, FeZn_10_** **δ, FeZn_7_**	8–137–10 [58]	δ, FeZn_10_δ, FeZn_7_ [59]	7–11.510.87	δ_1_, FeZn_10_	8.1–13.4
**ζ, FeZn_13_**	6–7	ζ, FeZn_13_	5–6	ζ, FeZn_13_	6.5–7.5
**η, Zn(Fe)**	0.04	η, Zn(Fe)	0	η, Zn(Fe)	0.03

**Table 4 materials-15-05887-t004:** Values of the heat treatment parameters and corrosion resistance of zinc coating after HT.

Parameter of Heat Treatment		Time to Red Corrosion Appearance, Hours/Days
Temperature t, °C	Time τ, min.	
0	0	360/15
270	7	360/15
	9	360/15
	11	336/14
350	7	360/15
	9	336/14
	11	312/13
430	7	336/147
	9	312/13
	11	288/12

## Data Availability

Data is contained within the article.

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
