# Peer review of "The Influence of Heat Treatment on Corrosion Resistance and Microhardness of Hot-Dip Zinc Coating Deposited on Steel Bolts"

_materials, 2022, doi:10.3390/ma15175887_

Round 1
Reviewer 1 Report
1. The subject covered in the article falls well within the scope of the journal, but the number of words outside the reference is less than 5000. It is therefore a little below the standards for this type of article, even if "officially" the journal does not limit the size/length.
2. - Title: It is not explicit enough for a title. The author investigated the evolution of micro-hardness or the influence of the heat treatment on the micro-hardness. Therefore, it is better to mention the micro-hardness for the title. The article should show exactly what it focuses on by reading the title. Maybe instead indicate the ‘the influence of the heat treatment on the corrosion resistance and micro-hardness of hot dip zinc coating deposited on steel bolts’?
3. - Introduction: the authors introduced and compared various galvanizing methods. But the authors insisted too much on this so that the readers would expect the results in this direction. The corrosion resistance mechanisms, microstructure evolution and the optimization of the heat treatment of the hot dip zinc coatings should be detailed in this part.
4. - Last paragraph of introduction: this is where the authors have to introduce everything that is going to be done and presented later in the paper and there are only 3 lines. It is better to develop a little more the different parameters that the author are going to test, what the authors are going to focus the discussion on and why.
5. - Results : The authors stated that the increase in hardness was caused by the increase in the thickness of iron-rich layers. It is also necessary to illustrate how and why these layers evolute at various HT conditions. In addition, the autors compared the Fe content of coatings which was heat treated and without HT. It is realy surprising that the authors obtained these precise values (1.72, 7.31 wg.%). Well then, readers would want to know how the authors measured the Fe content and what the uncertainties/precisions are when calculating the Fe content?
6. There is a big differnce in microstructure between Fig. 2b and Fig.2c. It is necesary to detail the difference and give an explaniation on the microstructre evolution during HT. P.S. The EDS results shown in Fig. 2b and Fig. 2c are difficult to analyze due to their low resolution and small size. What are the relative proportions of phaes in the coatings heat treaed at various conditions?
7. It is a good way to prodict the hardness or the corrosion resistance of the coatings by using equations. However, It is better to give the parameters' effective ranges and the stiffness of the expressions. For example,
yhv = -192. 39, if t = 0 and τ =0 (Eq. 1), however,the HV0.02 of the coating without HT is ~ 52. As for the stiffness of the equations (Eq. 1 and Eq. 2), it is better to verify the reliability and stability.
8. It is better to give a clear explanation on the reduction of corrosion resistance of the coatings treated in different conditions, as compared in Fig. 5.
9. Specify the physical properties with their unit of the axes in Fig. 8, please. For example HV0.02 for X and corrosion resistance for Y.
10. Regarding the applications of coated boltes, it is also interesting to know the changes in binding force for coatings treated in different conditions. Could the authors show some results in this, please?
11. It is difficult to understand the conclusions (3 and 4) stated by the authors. It seems we are facing with two contradictiory statements as compared between the conlusions (3, 4) and Figures 5, 6 and 8.
12. The manuscript needs careful editing (figures, content, grammers and etd.) before resubmission.
Author Response
Dear Reviewer,
See enclosed file.

Reviewer 2 Report
In this paper, the authors have studied the influence of heat treatment on the corrosion resistance of hot dip zinc coating. The authors have found that the controlled heat treatment can increase the hardness of this zinc coating for their applications. Therese results are very interesting. It is acceptable for publication in this journal after revision.
1).The abstract should be well simplified and summarized. The authors should be given deeply explain for the change of catalytic performances.
2).In this paper, the authors mainly consider the correlation between heat treatment and corrosion resistance. Therefore, what the evidence for the choose of the heat treatment temperature?
3).In table 1, if the authors considered the heat treatment temperature, I think that three heat treatment temperatures are difficult to reflect the change of corrosion resistance for this coating. The authors should be added two or more heat treatment temperature.
4).In introduction, the research background should be well summarized. For oxidation behavior of high-temperature materials or coating, the authors should be cited these references: Ceram Int. 2022;48:11518-11526. Mat Sci Eng B-solid. 2020;259:114580.
Ceram Int. 2018;44:19583-19589. Appl Surf Sci. 2022;591:153168. Vacuum. 2020;172:109067. Ceram Int. 2020;46:6698-6702. Mater Res Bull. 2018;107:484-491. Chem Phys Lett. 2018;698:211-217.
5).In table 2 and 3, what the evidence for the choose of Fe-Zn phases? In Table 2, the authors list the FeZn7 phase. Why not the authors consider the FeZn7 phase in Table 3?
6).In Figure 2, how to affirm the existence of these Fe-Zn phases? The author should be provided the XRD.
7). In Figure 7, why the heat treatment temperature can improve the corrosion behavior of this coating? Please deeply explained.
Author Response
Dear Reviewer,
See enclosed file

Reviewer 3 Report
Journal: Materials (ISSN 1996-1944)
Manuscript ID: materials-1867309
Type: Article
Title: The Influence of the Heat Treatment on the Corrosion Resistance of Hot Dip Zinc Coating Deposited on Steel Bolts.
Authors: Dariusz Jędrzejczyk*, Elżbieta Szatkowska.
a) Figure 1: remove the background (horizontal lines) from it and put the legend inside it verically?
b) Figure 2: why the author didn’t measure the distribution of the particles from SEM images and the average particle size?
c) Line 260: R square=0.99 how? Could the author add more detail about this value (correction rate) numerically using Figure 1 using the software.
d) Line 321: The same note in (c).
e) Why the author did not utilize XRD diffraction to construct the phases of the materials formed?
f) Figure 7 and Figure 8: the same note in (a).
g) For references, choose recent refs. Please, refer to these refs. EDS analysis
DOI: https://doi.org/10.1088/1742-6596/1795/1/012059
DOI: https://hal.archives-ouvertes.fr/hal-02443179
Best Regards
Author Response
Dear Reviewer,
See enclosed file

Round 2
Reviewer 1 Report
Accept in present form.
Reviewer 2 Report
The authors have made required changes. I recommend it for publication in its current form.